# The Impact of Gestational Weight Gain on the Risks of Adverse Maternal and Infant Outcomes among Normal BMI Women with High Triglyceride Levels during Early Pregnancy

**DOI:** 10.3390/nu13103454

**Published:** 2021-09-29

**Authors:** Xia-Fei Jiang, Hui Wang, Dan-Dan Wu, Jian-Lin Zhang, Ling Gao, Lei Chen, Jian Zhang, Jian-Xia Fan, He-Feng Huang, Yan-Ting Wu, Xian-Hua Lin

**Affiliations:** 1The International Peace Maternity and Child Health Hospital, School of Medicine, Shanghai Jiao Tong University, Shanghai 200030, China; jxf0724@foxmail.com (X.-F.J.); woodendenny@163.com (D.-D.W.); cl_pop@163.com (L.C.); ipmch@foxmail.com (J.Z.); fanjianxia122@126.com (J.-X.F.); 2Shanghai Key Laboratory of Embryo Original Diseases, Shanghai 200030, China; 3Department of Obstetrics and Gynecology, Maternity and Child Health Hospital of Songjiang District, Shanghai 201620, China; wanghuilinzifu@126.com; 4Obstetrics and Gynecology Hospital, Institute of Reproduction and Development, Fudan University, Shanghai 200011, China; zjl0039@163.com (J.-L.Z.); gaoling6633@126.com (L.G.)

**Keywords:** gestational weight gain, maternal triglyceride, early pregnancy, maternal and infant outcomes

## Abstract

A high maternal triglyceride (mTG) level during early pregnancy is linked to adverse pregnancy outcomes, but the use of specific interventions has been met with limited success. A retrospective cohort study was designed to investigate the impact of gestational weight gain (GWG) on the relationship between high levels of mTG and adverse pregnancy outcomes in normal early pregnancy body mass index (BMI) women. The patients included 39,665 women with normal BMI who had a singleton pregnancy and underwent serum lipids screening during early pregnancy. The main outcomes were adverse pregnancy outcomes, including gestational hypertension, preeclampsia, gestational diabetes, cesarean delivery, preterm birth, and large or small size for gestational age (LGA or SGA) at birth. As a result, the high mTG (≥2.05mM) group had increased risks for gestational hypertension ((Adjusted odds ratio (AOR), 1.80; 95% CI, 1.46 to 2.24)), preeclampsia (1.70; 1.38 to 2.11), gestational diabetes (2.50; 2.26 to 2.76), cesarean delivery (1.22; 1.13 to 1.32), preterm birth (1.42, 1.21 to 1.66), and LGA (1.49, 1.33 to 1.68) compared to the low mTG group, after adjustment for potential confounding factors. Additionally, the risks of any adverse outcome were higher in each GWG subgroup among women with high mTG than those in the low mTG group. High mTG augmented risks of gestational hypertension, preeclampsia, preterm birth, and LGA among women with 50th or greater percentile of GWG. Interestingly, among women who gained less than the 50th percentile of GWG subgroups, there was no relationship between high mTG level and risks for those pregnancy outcomes when compared to low mTG women. Therefore, weight control and staying below 50th centile of the suggested GWG according to gestational age can diminish the increased risks of adverse pregnancy outcomes caused by high mTG during early pregnancy.

## 1. Introduction

The accumulation of lipids in maternal metabolism, as well as the development of maternal hyperlipidemia, are essential characteristics of maternal lipidomic variation during pregnancy, and maternal lipid metabolism has an essential role in fetal growth and late pregnancy outcomes. Furthermore, maternal blood triglyceride (mTG) is a critical energy supplier, and its concentration alters to enhance the utilization of substrates critical for fetal growth and development during pregnancy [1].

Triglyceride levels increase in all pregnancies regardless of BMI as a normal physiological function of pregnancy [2,3,4,5]. It has been shown that the increase of mTG is not always dependent on maternal BMI [6]. However, high levels of serum total cholesterol (TC) and/or triglyceride (TG) during the first trimester have been closely associated with greater risks of preterm birth, LGA, preeclampsia, gestational hypertension [7,8,9,10]. The majority of pregnant women—even those with normal BMI—are at high risk for elevated triglyceride levels; therefore, they are at an increased risk for adverse maternal and fetal outcomes. 

Studies show that personalized weight management during pregnancy results in a decreased risk of perinatal complications and that it is an effective intervention strategy for the prevention of adverse pregnancy outcomes [11,12,13]. As an indirect reflection of the nutrition supply during the entire gestation period, gestational weight gain (GWG) is intricately linked to lifestyle factors. Insufficient or excessive GWG has been linked with adverse pregnancy outcomes [14]. Accordingly, the range of weight gain for pregnant women has attracted immense research interest, and awareness of the importance of GWG control is increasing. In 1970, guidelines regarding GWG developed by The Institute of Medicine (IOM) recommended clinical decision-making for weight control, and these guidelines were subsequently revised in 1990 and 2009 [14,15]. In 2016, the standards generated from healthy, well-nourished women were used to provide guidelines on appropriate GWG range worldwide [16]. In 2019, a meta-analysis published in JAMA reported another reference range in which the amount of GWG was defined in the setting of fewer adverse pregnancy outcomes [17]. 

Following previous analyses, the current evidence surrounding the relationship between GWG and maternal lipid profile shows that maternal GWG affects maternal, but not fetal lipid, profile differently from pre-pregnancy BMI [18]. Hence, we hypothesized that GWG control may also decrease adverse birth outcomes when hyperlipidemia occurs in early pregnancy. Since there is no current consensus on what constitutes effective intervention for hyperlipidemia in women with normal pre-pregnancy BMI, we conducted an observational study as a first attempt to fill this gap by collecting GWG data from normal-weight females with dyslipidemia in early pregnancy and examined the correlation between high mTG and adverse pregnancy outcomes. In addition, we also determined the recommended reference ranges and percentiles needed to reduce rates of complications in the perinatal phase for women with normal early pregnancy BMI when hyperlipidemia occurs in early pregnancy.

## 2. Materials and Methods 

### 2.1. Study Population

This cohort study utilized the data available in the electronic medical record system of Shanghai International Peace Maternal Health Hospital (IPMCH) and was based on in-hospital deliveries from Jan 2013 to Jun 2016 (Figure 1). The study population was based on the data published in the journal JCEM, in 2019 [10]. During the study period, a total of 50486 births were recorded in this hospital. Exclusion criteria were as follows: (1) cases with missing data for BMI of early pregnancy; (2) women with BMI < 18.5 or ≥ 25 kg/m^2^. In the end, 39,665 singleton pregnancies were included in the study. Ethics approval was obtained from the research ethics committee at IPMCH and written informed consent was provided by participants before study participation.

### 2.2. Data Collection and Independent Variables

The hospital’s electronic medical records collect data on each birth delivered in the hospital. For each delivery, standardized forms were used for antenatal, obstetric, and neonatal care. Maternal demographic, anthropometric characteristics, and reproductive history were collected at the first prenatal visit between 9 and 14 weeks gestation. 

Gestational age estimated in weeks was ascertained by the dating method using the last menstrual period (LMP) and tracked by ultrasound at the first antenatal visit. We calculated the length of gestation based on the ultrasound if the disagreement between the two methods was over one week.

The difference between measured weight shortly before delivery and measured weight in the first trimester (between 9 and 14 weeks gestation) was calculated as the total maternal GWG throughout gestation. Women were stratified based on delivery gestational age and determined to belong in the 3rd, 10th, 25th, 50th, 75th, 90th, and 97th percentile, in accordance with GWG values provided from normal BMI women involved in the INTERGROWTH-21st Project [16]. 

Clinical data were collected at the pregnant women’s first antenatal visit before 14 weeks gestation. Blood was drawn at the specified intervals between 7:00 a.m. and 9:00 a.m. Serum triglyceride concentrations were determined with an enzymatic assay (Roche Diagnostics, Mannheim, Germany) and a Cobas c702 analyzer. The inter-assay coefficient of variation was < 2.5% for triglycerides. The cut-off point of mTG at the 90th (2.05 mM) percentile was defined according to a previous study [10]. We defined values falling below the 90th percentile as “low mTG”, and values equal to or above the 90th percentile as “high mTG”.

### 2.3. Definition of Outcomes

The diagnoses were coded according to the 9th or 10th revision of the International Statistical Classification of Diseases (ICD-9 or ICD-10). Briefly, preterm birth was indicated by spontaneous birth before 37 weeks gestation [19]. Gestational hypertension was defined as new-onset hypertension with resting blood pressure ≥ 140/90 mmHg after the 20th gestational week in previously normotensive women. Pre-eclampsia (PE) was defined as gestational hypertension accompanied by proteinuria occurring after the 20th gestational week. Preeclampsia cases were not included in the gestational hypertension analysis and vice versa. A standard 75-g oral glucose tolerance test (OGTT) was performed, and plasma glucose was measured at 1 and 2 h after a 10 h overnight fast at 24−28 gestation weeks in women without a history of diabetes. According to the IADPSG GDM diagnostic criteria, GDM was diagnosed when any of the the following criteria was met or exceeded in the 75-g OGTT: 1) Fasting plasma glucose value: 92 mg/dL (5.1 mmol/L); 2) 1-h plasma glucose value: 180 mg/dL (10.0 mmol/L); 3) 2-h plasma glucose value: 153 mg/dL (8.5 mmol/L) [20,21,22]. Small and large for gestational age were defined as an infant with a birth weight below the 10th percentile and above the 90th percentile, respectively. The main outcome used in the analysis was the following adverse outcomes: gestational hypertension, preeclampsia, gestational diabetes, cesarean delivery, preterm birth, and large or small size for gestational age at birth. Any adverse outcome was defined as the presence of at least 1 of the previous outcomes. 

## 3. Statistical Analysis

GWG, mTG levels, and maternal early pregnancy BMI are presented as median and quartile 1 and quartile 3. Categorical variables were represented as frequencies with proportions. We applied multilevel linear regression models to assess associations between maternal baseline characteristics and total gestational weight gain among mTG groups. The absolute risk for any adverse outcome was estimated across the full range of early pregnancy mTG and gestational weight gain. Absolute risks were calculated as the percentage of women with any adverse outcome within each combination of mTG and GWG categories. Similarly, the absolute risks were estimated for any adverse outcome and individual pregnancy outcomes across the range of GWG categories within each mTG group. Univariate logistic regressions were used to estimate odds ratios (ORs) with a 95% confidence interval (CI). The ORs for any adverse outcome were calculated for low mTG and high mTG women within each GWG category.

Multivariable analyses were adjusted for potential confounders. Maternal age (≤24, 25–29, 30–34, or ≥35 years), height (≤154, 155–164, 165–174, ≥175 cm), parity (once, twice, or more than three times), education (≤9, 10–12, 13–15, or ≥16 years), and maternal residential status (resident or immigrant) were included as covariates and to assess the connection between mTG levels in early pregnancy and incidences of adverse outcome.

Absolute risk difference compared to normal mTG was calculated as the difference between the risk of any adverse outcome in the high mTG group and the risk of that event in the low mTG group. To reduce the bias caused by fewer samples, women were combined into two subgroups as follows: < 3rd, 3rd–10th, and 10–25th as <25th subgroup and 75–90th, 90–97th, and ≥97th as ≥75th subgroup. The confidence interval was obtained with standard statistical packages [23]. A confidence interval that contained a zero meant that there was no significant difference between the event and the control in terms of risk.

All statistical analyses were performed using SPSS package version 16.0 (SPSS Inc., Chicago, IL, USA). A two-tailed *P*-value of less than 0.05 was used as the threshold for statistical significance. 

## 4. Results

### 4.1. Baseline Characteristics of the Study Population 

Women (*n* = 50,486) with a mean (±SD) age of 30.08 (±3.71) years were enrolled in the study. After excluding the women with BMI < 18.5 and ≥ 25 kg/m^2^, 39,665 women with complete GWG data were subgrouped into low (*n* = 36,153), and high (*n* = 3512) mTG groups (Figure 1). There were 39,665 women with normal early pregnancy BMI involved in the validation cohort as a result of sample availability. They had a median GWG of 14.1 kg (quartile 1 and 3, 11.1 and 17.1 kg), a median mTG of 1.20 mM (quartile 1 and 3, 0.95 and 1.55 mM), and a median early pregnancy BMI of 21.4 kg/m^2^ (quartile 1 and 3, 20.2 and 22.8 kg/m^2^). Women who were shorter, older, and multiparous were more likely to have high mTG during early pregnancy, and the factors of educational background (lower education), place of birth (immigrant), and ethnicity (non-Han Chinese ) could also contribute to the lipid profile (Table 1).

### 4.2. Maternal TG in Early Pregnancy and Risks of Adverse Maternal and Neonatal Outcomes

Any adverse outcome occurred in 22,637 women (57.1%), ranging from 56.0% (20,255 of 36.153) among women categorized as low mTG to 67.8% (2382 of 3512) among women categorized as high mTG (Table 2).

Compared with the low mTG group, the high mTG group had increased levels of risk of the following adverse pregnancy outcomes: gestational hypertension, preeclampsia, gestational diabetes, cesarean delivery, postpartum hemorrhage, preterm birth, LGA, and macrosomia after adjusting for maternal age, height, education, place of birth, and parity. The rates and AORs were as follows: gestational hypertension (AOR, 1.80; 95% CI, 1.46 to 2.24); preeclampsia (1.70; 1.38 to 2.11); gestational diabetes (2.50; 2.26 to 2.76); cesarean delivery (1.22; 1.13 to 1.32); postpartum hemorrhage (1.50; 1.14 to 1.99); preterm birth (1.42, 1.21 to 1.66); LGA (1.49, 1.33 to 1.68); macrosomia (1.50, 1.29 to 1.74); and any adverse outcome (1.54; 1.42 to 1.67) (Table 2). Slightly higher rates and risks of neonatal intensive care unit admission (NICU) were also found in the high mTG group compared to the low mTG group. At the same time, high mTG was associated with a slightly decreased risk of SGA. However, the two groups showed similar rates of intrahepatic cholestasis of pregnancy ( ICP), placental abruption, placenta previa, and low birth weight (Table 2). 

### 4.3. The Absolute Risk for Any Adverse Outcome Categorized by Maternal Early Pregnancy TG Level in Gestational Weight Gain (GWG) Percentiles

For gestational hypertension, preeclampsia, and LGA, the curves represented trends of higher absolute risk for GWG ≥ 25th in the high mTG group when compared with the low mTG group (Figure 2A,B,F). The absolute risks for cesarean delivery were found to be higher in nearly all percentiles of GWG except less than 3rd in the high mTG group when compared with the low mTG group (Figure 2D). For preterm birth, trends of higher absolute risk for GWG ≥ 50th were shown in the high mTG group when compared with the low mTG group (Figure 2E). High mTG women presented lower risks of SGA (Figure 2G) and showed a downward trend in the two groups with increasing GWG categories. For gestational diabetes and any adverse outcome, the absolute risks were found to be higher in all GWG percentiles subgroup among women with high mTG than those in the low mTG group (Figure 2C,H).

### 4.4. High mTG and the Risk of Any Adverse Outcome in Different GWG Groups in Women with Normal Early Pregnancy BMI

We further analyzed the association between mTG and the risks of any adverse outcome in four subgroups of women with different GWG (Figure 3). Women with high mTG had higher risks for both gestational hypertension and preeclampsia vs. women with low mTG in gestational weight gain of the 50 to 75th percentile (gestational hypertension: 3.3% vs. 1.7%, respectively; absolute risk difference, 1.6% (95% CI, 0.4% to 2.7%); AOR, 2.16 (95% CI, 1.34 to 3.26); preeclampsia: 3.9% vs. 1.8%, respectively; absolute risk difference, 2.1% (95% CI, 0.8% to 3.4%); AOR, 2.11 (95% CI, 1.39 to 3.20)), and among women with gestational weight gain ≥ 75th percentile (gestational hypertension: 4.5% vs. 2.6%, respectively; absolute risk difference, 1.9% (95% CI, 0.3% to 3.4%); AOR, 1.81 (95% CI, 1.32 to 2.47); preeclampsia:5.6% vs. 3.8%, respectively; absolute risk difference, 1.8% (95% CI, 0.1% to 3.5%); AOR, 1.64 (95% CI, 1.25 to 2.16)). Interestingly, we found that high mTG level had no correlation to higher risks of gestational hypertension and preeclampsia among pregnant females with gestational weight gain <25th and 25th to 50th percentiles. 

Furthermore, the risks of preterm birth and LGA were higher in the high mTG group among women vs. the low mTG group among women with gestational weight gain of 50th to 75th percentile (preterm birth: 6.7% vs. 4.1%, respectively; absolute risk difference, 2.6% (95% CI, 0.9% to 4.2%); AOR, 1.48 (95% CI, 1.10 to 2.01); LGA: 13.1% vs. 7.9%, respectively; absolute risk difference, 5.2% (95% CI, 3.0% to 7.4%); AOR, 1.73 (95% CI, 1.38 to 2.17)), and among pregnant females with gestational weight gain of ≥ 75th centile (preterm birth: 8.4% vs. 5.4%; absolute risk difference, 3.0% (95% CI, 0.9% to 5.1%); AOR, 1.70 (95% CI, 1.36 to 2.13); LGA: 19.0% vs. 14.0%; absolute risk difference, 5.0% (95% CI, 2.0% to 8.0%); AOR, 1.45 (95% CI, 1.23 to 1.71)). Among women with gestational weight gain <25th and 25th to 50th centiles. No statistically significant difference was found in the rate of preterm births between high and low mTG women.

High mTG was associated with a higher incidence of cesarean delivery among pregnant females with a gestational weight gain in the 50th percentile or greater subgroups. The risk of cesarean delivery was consistently higher among high mTG women than among those in the normal mTG group, while the risks for GDM in the high-mTG group were all higher in each GWG centile subgroup. A slightly lower risk of SGA with a smaller absolute risk difference were found in high mTG compared to those in low mTG women.

## 5. Discussion

In this analysis of 39,665 pregnancies from a large retrospective cohort study, high mTG (≥2.05 mM) showed increased risks for gestational hypertension, preeclampsia, gestational diabetes, cesarean delivery, preterm birth, and LGA, compared to the low mTG group. Additionally, the risks of any adverse pregnancy outcome were higher in any GWG subgroup based on the weight range recommended by INTERGROWTH-21st guidelines among women with high mTG vs. women with low mTG. Further analysis showed that high mTG increased the risks of adverse pregnancy outcomes among women in the ≥50th percentile GWG subgroups, which did not happen among normal early pregnancy BMI women with less than the 50th percentile of GWG.

To our knowledge, this is the first study dedicated to providing an suggested range of gestational weight gain for women with normal weight but with abnormally high mTG level rather than rough outpatient follow-up without specific intervention. The methodology for GWG control proved to be a robust and efficient intervention strategy for the prevention of adverse outcomes. Consistent with a recent meta-analysis published in JAMA [17], the present study used GWG ranges that had been associated with a lower risk for any adverse outcome to estimate optimal GWG ranges. However, like the IOM guidelines, the meta-analysis can not be applied to specific features of the Chinese population, of which the majority consists of normal prepregnancy-weight women. Consistent with the INTERGROWTH-21st guidelines [16], the present study used smoothed GWG centiles for women of normal weight according to gestational age to determine optimal GWG ranges (The reference table can be obtained via the link as follow, https://www.ncbi.nlm.nih.gov/pmc/articles/PMC4770850/table/tbl3/?report=objectonly), which means that more specific and explicit clinical management can be undertaken. However, the INTERGROWTH-21st guidelines are inefficient to provide optimal ranges when high mTG is common in the pregnant population. Addressing these deficiencies, we have provided practical data, as well as an accompanying percentile GWG chart for normal BMI women with high mTG, which can provide week-by-week guidance, based on absolute risks, providing clear quantitative information to clinicians and patients.

Weight gain beyond recommended levels is termed excessive GWG and has been linked to fetal and maternal complications, such as LGA and postpartum weight retention. Accordingly, limiting GWG in women is a significant factor in the prevention of main adverse pregnancy outcomes. Serum TG levels increase significantly from early to late pregnancy, due to hormonal changes and maternal lipid metabolism. In a previous study, we were able to demonstrate that high mTG levels occurring in early pregnancy, a risk factor independent of pregnancy BMI category, are associated with an increased incidence of preterm delivery [10]. Our study further confirms that dyslipidemia is closely related to adverse pregnancy outcomes. The absolute risk corresponding to these outcomes is statistically significant compared with women of low mTG group, after eliminating all potential confounding factors. Combining the absolute risk of high and low blood lipids, we have identified a trend showing that the greater the percentile of GWG, the higher the value of absolute risk in the vast majority of adverse outcomes. Data from observational analyses associated with preterm labor and LGA indicate a significant influence of weight control on blood lipids. Notably, however, the relationship between the incidence of gestational diabetes and GWG is not straightforward, in contrast to other outcomes. This may be related to care preferences in a Chinese hospital. In China, regular nutrition intervention is performed after the diagnosis of GDM after 20 weeks of gestation, which mainly focuses on weight control and dietary counseling [24]. Furthermore, pregnant women with GDM receive greater attention and more intervention when they also have high triglycerides. Previous studies give differing and controversial views regarding the risk of GDM in normal-weight females and overweight females [25,26,27]. This may be because the outcomes are also influenced by nutritional counseling and dietary adjustments and cannot be accounted for by weight gain alone [28]. In addition, the low rate of smoking in pregnant women may cause few differences between low and high mTG groups, which will result in a bias in the analyses. Therefore, we did not adjust the factor of smoking as a potential confounding factor.

Excessive maternal GWG is regarded as an important contributor to several adverse maternal and neonatal outcomes [29,30,31,32]. Guidelines for how much weight women should gain during pregnancy were made according to early pregnancy BMI [14], regardless of maternal lipid profiles. It is known that most of the plasma triglyceride of humans comes from food intake. Despite the fact that TG levels increase in all pregnancies as a normal physiological function, increased mTG level in early pregnancy suggests an excess intake of fat, even if BMI is normal. While weight increases as the pregnancy progress, the speed of adipose tissue accumulation differs in women, depending on the amount of fat intake [33]. Moreover, weight gain is not linear during pregnancy. In our study, we referred to the standard GWG corresponding to gestational age advised by the Fetal Growth Longitudinal Study of the INTERGROWTH-21st project, which was a longitudinal cohort study performed in eight geographically diverse urban regions, including China [16]. We assessed the risks of adverse pregnancy outcomes in low and high mTG groups and evaluated the risk differences between them. Elevated TG levels led to significantly higher risks of gestational hypertension, preeclampsia, preterm birth, and LGA in normal BMI females with GWG of ≥ 50th percentile, while the risks attenuated among females with GWG < 50th percentile. These results strongly suggest that limiting gestational weight gain could likely prevent major adverse maternal and infant outcomes contributed to by high mTG during early pregnancy. Elevated mTG not only promotes the growth of the fetus [34], but has also been reported to enhance peroxidation and reduce antioxidant capacity in the placenta. Both placental oxidative stress and large size for the gestational age of the fetus are risk factors for gestational hypertension and preterm birth [35,36]. Additional studies should examine the benefit of modifying maternal fat intake in pregnancy and GWG control in females with high mTG during early pregnancy. 

The major strengths of the present study are a large sample size, hospital-based samples, and a homogeneous ethnic population. However, our study also has some limitations. First, this research did not use a country-wide cohort, which means that the limited samples from one hospital may have reduced the validity of the conclusions. Second, we took maternal blood samples only once in the first gestational period, so it was not possible to track both the trajectory of TG levels and changes occurring in those levels in the low GWG women with favorable outcomes. Thirdly, the optimal weight gain during pregnancy is individual to each woman and determined by many factors. Among them, nutrition and physical activity can significantly affect weight gain in pregnancy. Thus, research including the nutrition and physical exercise assessment should be taken to deepen the understanding of GWG. 

## 6. Conclusions

Our results indicate that high maternal TG levels during the first gestational period are associated with a high incidence of adverse pregnancy outcomes, both in maternal and fetal aspects for females with normal BMI. Elevated mTG levels decreased the risks of main adverse maternal and infant outcomes in women who experienced controlled gestational weight gain, suggesting that limiting weight gain in pregnant females with high TG could be a potential measure to reduce the occurrence of adverse pregnancy outcomes. GWG below the 50th percentile based on INTERGROWTH-21st guidelines should be recommended for effective prevention to diminish the increased risks of adverse pregnancy outcomes caused by high mTG during early pregnancy. Additional studies to confirm that weight control in an appropriate range for hyperlipidemia in pregnant women is an effective measure to reduce adverse pregnancy outcomes should be considered in the future, including prospective cohorts and/or randomized controlled trials. 

## Figures and Tables

**Figure 1 nutrients-13-03454-f001:**
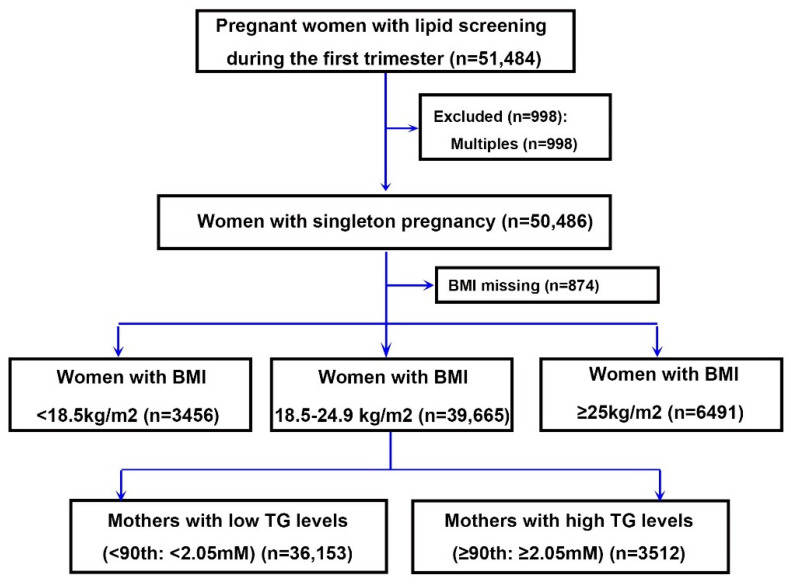
Flowchart for selecting women included in this study.

**Figure 2 nutrients-13-03454-f002:**
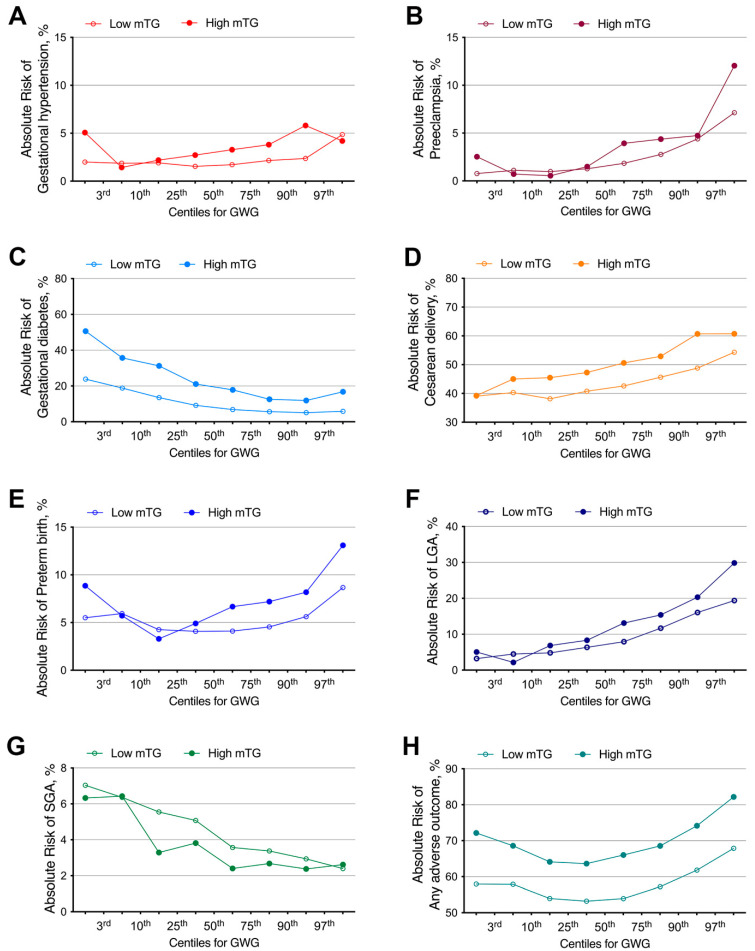
Association of gestational weight gain (GWG) percentiles with adverse pregnancy outcomes in low mTG and high mTG groups. Absolute risks of adverse pregnancy outcomes in different gestational weight gain (GWG) percentiles between low and high mTG group (A) Absolute risks of gestational hypertension; (**B**) Absolute risks of preeclampsia; (**C**) Absolute risks of gestational diabetes; (**D**) Absolute risks of cesarean delivery; (**E**) Absolute risks of preterm birth; (**F**) Absolute risks of LGA; (**G**) Absolute risks of SGA; (**H**) Absolute risks of any adverse outcome.

**Figure 3 nutrients-13-03454-f003:**
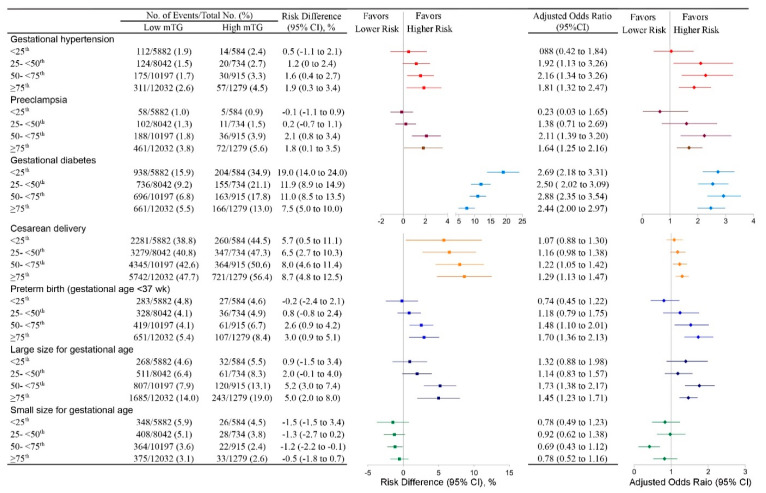
Association of maternal TG levels with adverse pregnancy outcomes among normal BMI women stratified by gestational weight gain. Absolute risk differences were calculated as the difference between the risks of high mTG and low mTG groups in each category of gestational weight gain (GWG). Adjusted odds ratios were shown for the association of mTG with adverse pregnancy outcomes and all analyses were adjusted for age, height, education, parity, place of birth, and timing of the blood draw during gestation. The reference group was women with low mTG in each outcome and each GWG category. GWG categories of Normal BMI women were stratified by the threshold of 3rd, 10th, 25th, 50th, 75th, 90th, and 97th percentile based on delivery gestational age. In accordance with the INTERGROWTH-21st Project, subgroups of <3rd, 3–10th, 10–25th were combined as <25th subgroup, and subgroups of 75–90th, 90–97th, ≥97th were combined as ≥ 75th subgroup to reduce the bias caused by fewer samples. For each outcome, the total number of women in the studies represented by the sample size assessed the outcomes. Risk differences, adjusted odds ratios, and each corresponding 95% CIs were calculated from the results of multivariable models and adjusted for baseline risk imprecision.

**Table 1 nutrients-13-03454-t001:** Characteristics of study population.

	Entire Population		Low mTG		High mTG		*p* Value
	*n* = 39,665		*n* = 36,153		*n* = 3512		
Total gestational weight gain, median (q1 and q3), kg	14.10 (11.10 and 17.10)		14.10 (11.50 and 17.10)		14.20 (11.3 and 17.5)		
Maternal TG level, median (q1 and q3), mM	1.20 (0.95 and 1.55)		1.15 (0.92 and 1.44)		2.40 (2.20 and 2.76)		
Early pregnancy BMI ^a^, median (q1 and q3), kg/m^2^	21.42 (20.16 and 22.80)		21.34 (20.13 and 22.74)		22.40 (20.79 and 23.40)		
Maternal age (yr), No. (%)							<0.001
≤24	1552	(3.91)	1480	(4.09)	72	(2.05)	
25–29	17,590	(44.35)	16,381	(45.31)	1209	(34.42)	
30–34	15,709	(39.60)	14,143	(39.12)	1566	(44.59)	
≥35	4814	(12.14)	4149	(11.48)	665	(18.94)	
Height, No. (%)							<0.001
≤154	2048	(5.16)	1851	(5.12)	197	(5.61)	
155–164	24,695	(62.26)	22,395	(61.95)	2298	(65.43)	
165–174	12,462	(31.42)	11,483	(31.76)	981	(27.93)	
≥175	460	(1.16)	424	(1.17)	36	(1.03)	
Education (years), No. (%)							<0.001
≤9	790	(1.99)	677	(1.87)	113	(3.22)	
10–12	2364	(5.96)	2117	(5.86)	247	(7.03)	
13–15	8439	(21.28)	7646	(21.15)	793	(22.58)	
≥16	27,538	(69.43)	25,229	(69.78)	2309	(65.75)	
Missing	534	(1.35)	484	(1.34)	50	(1.42)	
Birth place, No. (%)							<0.001
Residents	29,690	(74.85)	27,243	(75.35)	2447	(69.68)	
Immigrants	9975	(25.15)	8910	(24.65)	1065	(30.32)	
Parity, No. (%)							<0.001
1	32,614	(82.22)	29,998	(82.98)	2616	(74.49)	
2	6848	(17.26)	5988	(16.56)	860	(24.49)	
≥3	203	(0.51)	167	(0.46)	36	(1.03)	
Ethnicity, No. (%)							<0.001
Han Chinese	39,080	(98.53)	35,635	(98.57)	3446	(98.12)	
Other	585	(1.47)	518	(1.43)	66	(1.88)	

^a^ Calculated as weight in kilograms divided by height in meters squared.

**Table 2 nutrients-13-03454-t002:** Maternal TG in early pregnancy and risks of adverse pregnancy outcomes.

	Entire Population		mTG 10–90th			mTG > 90th		
	*n* = 39,665		*n* = 36,153			*n* = 3512		
Types of adverse outcomes, No. (%), AOR (95% CI)								
Gestational hypertension	843	(2.13)	722	(2.00)	1.00 (ref)	121	(3.45)	1.80 (.46, 2.24)
Preeclampsia	933	(2.35)	809	(2.24)	1.00 (ref)	124	(3.53)	1.70 (1.38, 2.11)
Gestational diabetes	3719	(9.38)	3031	(8.38)	1.00 (ref)	688	(19.59)	2.50 (2.26, 2.76)
Cesarean delivery	17,438	(43.96)	15,647	(43.28)	1.00 (ref)	1791	(51.00)	1.22 (1.13, 1.32)
Placental abruption	25	(0.06)	22	(0.06)	1.00 (ref)	3	(0.09)	1.64 (0.48, 5.58)
Placenta previa	477	(1.20)	437	(1.21)	1.00 (ref)	40	(1.14)	0.84 (0.59, 1.19)
Intrahepatic Cholestasis of Pregnancy (ICP)	443	(1.12)	412	(1.14)	1.00 (ref)	31	(0.88)	0.67 (0.44, 1.03)
Postpartum hemorrhage	540	(1.36)	473	(1.31)	1.00 (ref)	67	(1.91)	1.50(1.14, 1.99)
Preterm birth	1912	(4.82)	1681	(4.65)	1.00 (ref)	231	(6.58)	1.42 (1.21, 1.66)
Small size for gestational age	1604	(4.04)	1495	(4.14)	1.00 (ref)	109	(3.10)	0.78 (0.63, 0.97)
Large size for gestational age	3727	(9.40)	3271	(9.05)	1.00 (ref)	456	(12.98)	1.49 (1.33, 1.68)
Birth weight								
Low birthweight (<2500 g)	1050	(2.65)	762	(2.11)	1.00 (ref)	101	(2.88)	1.10 (088, 1.39)
Macrosomia (>=4000 g)	2123	(5.35)	1863	(5.15)	1.00 (ref)	260	(7.40)	1.50 (1.29, 1.74)
NICU admission	3675	(9.27)	3297	(9.12)	1.00 (ref)	378	(10.76)	1.27 (1.13, 1.44)
Any adverse outcome	22,637	(57.07)	20,255	(56.03)	1.00 (ref)	2382	(67.82)	1.54 (1.42, 1.67)

Abbreviations: BMI, body mass index; OR, odds ratio; CI, confidence interval; NICU, neonatal intensive care unit. Data are shown as adjusted OR (95% CI), adjusted for maternal age, height, education, place of birth, parity. The rates of early preterm delivery were calculated by comparison with all other deliveries. The rates of late preterm delivery were calculated by comparison with deliveries at 34 weeks or later.

## Data Availability

The data presented in this study are available on request from the corresponding author.

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
