# Peer review of "The Impact of Gestational Weight Gain on the Risks of Adverse Maternal and Infant Outcomes among Normal BMI Women with High Triglyceride Levels during Early Pregnancy"

_nutrients, 2021, doi:10.3390/nu13103454_

Round 1

Reviewer 1 Report

Thank you. This is a very interesting study. I have no comments.

Congratulations

Good luck !

Author Response

We would like to show our great appreciation for your positive comments and taking  time to review our manuscript.

Reviewer 2 Report

This was a novel and interesting article exploring the relationship between early-pregnancy triglycerides, gestational weight gain and adverse pregnancy outcomes. The findings, especially those relating to how lower GWG < 50th centile may be able to attenuate risk of adverse pregnancy outcomes in those with raised TGs in early pregnancy are clinically important. I offer some suggestions below to refine your manuscript

Abstract:

Line 27 – rather than simply stating risk increased, it would be useful to include some data. For example, odds of any adverse outcome, or named outcomes such as gestational hypertension or GDM.

Line 27 – include definition of high mTG.

Introduction:

Lines 46-47 ‘Lipid levels in pregnant women with normal BMI are still above those in normal non pregnant women from the first to the third trimester’ needs rephrasing/simplifying just to state that TG levels increase in all pregnancies regardless of BMI as a normal physiological function of pregnancy.

Line 58 ‘GWG is intricately linked to lifestyle factors and it is an important factor for pregnant women’s overall BMI classification.’ Do you mean their BMI classification post pregnancy? GWG doesn’t influence BMI in pregnancy as it is usual to classify women by their pre-pregnancy BMI.

Lines 60-66 – how do these GWG recommendation vary? Important to mention that the 2009 IOM recommendation are specific to pre-pregnancy BMI.

You do not mention the current evidence surrounding the relationship between GWG and maternal lipid profile. There are studies examining this, although many are conducted in women with obesity. Nonetheless, you should address this in your introduction. See Sommer al: https://doi.org/10.1186/s12884-015-0512-5 or Scifres et al: https://doi.org/10.1002/oby.20576

Methods:

Line 99 – at what point in trimester 1 was weight measured? Are women weighed when they go into labour or at an antenatal visit in late pregnancy? Might this vary between women?

Line 109 – please include further information on the justification for the use of low and high mTG cut-offs.

Line 122 – you have used the IADPSG GDM diagnostic criteria, please state this.

Lines 125-127 – which population reference values did you use to define SGA and LGA?

Statistical analysis

Was smoking included as a confounder? This is an important factor that can influence risk for all of the adverse pregnancy outcomes you are examining. This important factor needs to be added to your analysis.

Results:

Table 1. Are the numbers in the brackets percentages? This should be labelled clearly.  It would be useful to perform independent t tests to see if there are any significant differences in baseline characteristics between low and high mTG groups.

Figure 2 – I’m not convinced of the value of figure 2A and B, as it’s difficult to visually compare the low vs high mTG groups on separate graphs. I suggest a separate graph for each outcome, with low and high mTG included.

Figure 3 – it needs to be clear that the percentiles in the far left column refer to GWG – they are not currently labelled. What is the reference group for these odds ratios?

Line 232 ‘while it was shocked.’ Rephrase as this doesn’t make sense. Perhaps to ‘Interestingly, there was no correlation…’

Discussion:

Lines 265-266. The opening statement ‘the absolute risk for any adverse outcome was the highest at both extremes of the GWG range.’ I don’t know quite what you mean, but this can’t be true as low GWG seems to attenuate risk of some adverse outcomes according to Figure 3.

Line 272-290. It would be useful to define the rate of GWG observed for each of the four centile groups. This will make it easier for those reading your paper to visualise the GWG. For your findings, can you make recommendations for GWG targets in order to attenuate risk of adverse outcomes following early high mTG?

Lines 300-301. You state you have eliminated all potential confounding factors. This isn’t the case as you don’t appear to have adjusted for smoking, nor have you considered maternal dietary intake or physical activity.

Line 319. You suggest that the increase in mTG in early pregnancy is due to excess intake of dietary fat. This may be a factor where mTG levels rise excessively; however, you should acknowledge that increase in mTGs is a normal physiological function of pregnancy.

Author Response

Thank you for the positive comments.

Reviewer 3 Report

Title: The Impact of Gestational Weight Gain on the Risks of Adverse Maternal and Infant Outcomes Among Normal BMI Women with High Triglyceride Levels During Early Pregnancy

The article focuses on the impact of gestational weight gain on the risks of adverse maternal and infant outcomes among pregnant women with normal BMI but high triglyceride level. The topic of the article is very important. However, there are some doubts about the methodology and interpretation of the study results:

  • The study is retrospective, observational and conducted on a non-representative sample. This type of research allows to observe relationships but cause and effect conclusions cannot be drawn. (257-262) High mTG also increased the incidence of cesarean delivery among pregnant females with a gestational weight gain in the 50th percentile or greater subgroups. The risk of cesarean delivery was consistently higher among high mTG women than among those in the normal mTG group, while the risk for GDM in the high-mTG group increased independently of gestational weight gain. High mTG slightly decreased the risks of SGA, with a smaller absolute risk difference.
  • In my opinion, it is not possible to recommend (Lines 34-36) “staying below 50% of the optimal gestational weight gain are recommended for the prevention of major adverse pregnancy outcomes in normal early” based on the study finding. Optimal weight gain during pregnancy is individual to each woman and determined by many factors. Caution should be taken in formulating such recommendations, as both excessive weight gain and insufficient weight gain can lead to complications in pregnancy. In addition, the study did not include an assessment of nutrition, which significantly affects weight gain in pregnancy. The conclusions drawn by the authors go beyond the conducted research (Lines 273-275) “To our knowledge, this is the first study dedicated to providing an exact range of gestational weight gain for clinical decision making rather than rough outpatient follow- up without specific intervention among women with normal weight but with abnormally high mTG level”.
  • The study population was divided into low mTD and high mTG according to triglyceride level. Please explain whether women with high TG levels before pregnancy were excluded from the study.
  • Data presented in descriptive form (lines 164-171; 184-188) are also included in the tables. Please consider shortening the data description in the text.
  • Please explain why the continuous data are presented as median and Q1 and Q3? The statistical analysis methodology states that tests were performed, but the results are not presented (lines 156-157) “A two-tailed P-value of less than 0.05 was used as the threshold for statistical
  • Please consider changing the phrase (line 232) While it was shocked to find that high  mTG level had no correlation to higher risks of gestational hypertension and preeclampsia among pregnant females with gestational weight gain < 25th and 25th to 50th percentiles
  • Lines 170-171 “Women with high mTG had increased rates of shorter, older, multiparous, lower-educated, immigrant, and non-Chinese Han women” - this sentence needs improvement.

In conclusion, this article requires major revisions before being published including English proofreading.

Author Response

Thank you for the positive comments.

Round 2

Reviewer 3 Report

I appreciate the effort the authors have put into improving the article. I accept the revised version.